# Activated Carbon Based on Recycled Epoxy Boards and Their Adsorption toward Methyl Orange

**DOI:** 10.3390/polym16121648

**Published:** 2024-06-11

**Authors:** Wenfeng Zhu, Jiacheng He, Qianxi Wang, Dongna Zhang, Guoquan Qi, Xuehua Cai, Peipei Li, Jiaoxia Zhang

**Affiliations:** 1State Key Laboratory of Oil and Gas Equipment, Tubular Goods Research Institute, China National Petroleum Corporation, Xi’an 710077, China; zhangdna@cnpc.com.cn (D.Z.); qiguoquan@cnpc.com.cn (G.Q.); caixuehua@cnpc.com.cn (X.C.); 2School of Materials Science and Engineering, Jiangsu University of Science and Technology, Zhenjiang 212003, China; hejc264270@163.com (J.H.); wqx2008137@163.com (Q.W.); 3School of Advanced Materials and Nanotechnology, Xidian University, Xi’an 710071, China; lip@xidian.edu.cn

**Keywords:** activated carbon, discarded epoxy resin, adsorption, methyl orange

## Abstract

With the swift progress of the electronics industry, discarded circuit boards have become an important source of non-degradable waste. In this work, discarded epoxy resin was collected as a precursor to prepare activated carbon (AC) through stepwise carbonization/activation methods. The rough carbon materials with a certain graphite and amorphous structure reveal the multiple oxygen-containing groups on their surface. In the process of studying the adsorption of methyl orange by activated carbon, it is found that the adsorption is in accordance with the quasi-secondary kinetic model, and equilibrium adsorption amounts can reach 41.051 mg/g. The adsorption isotherm of AC is more in line with the Langmuir model, and the saturation adsorption amount at three different temperatures is 23.137 mg/g, 30.358 mg/g, and 37.202 mg/g, respectively. The enthalpy (ΔH) is 17.30 KJ/mol in the adsorption process, which indicates that is a physical process with heat-absorbing capabilities. This work is of great significance with regard to the recycling of waste to reduce pollution and in terms of gaining economic benefits.

## 1. Introduction

With the rapid progress of electronic information and industrial technology, the performance requirements of electronic products are rapidly increasing, and the replacement frequency also has also risen sharply, which has led to the generation of a large amount of waste electrical and electronic equipment (WEEE) [1,2]. However, the annual global production of WEEE is growing at an alarming rate of 20 to 25 million tons per year, of which printed circuit boards (PCBs), which are an important part of all electronic equipment, make up 4–7% of the total mass of WEEE and are composed mainly of thermosetting epoxy resins [3,4]. Therefore, the recycling of epoxy resins has become a very important issue today. Thermosetting epoxy resins are three-dimensional crosslinked polymers with a complex chemical structure, which usually consists of epoxy groups and polyanhydrides containing aromatic or aliphatic structures, among others. This structure is difficult to be decomposed effectively in the recycling process and its insoluble and immiscible properties make it difficult to be recycled [5,6]. Secondly, epoxy resins inherently have excellent properties, such as high heat resistance and chemical resistance, which also make the recycling of epoxy resins difficult during the recycling process. At present, the most important treatment methods are incineration and direct landfill; however, incineration produces polluting harmful gases, causing air pollution and jeopardizing human health [7]. Direct landfill is not conducive to soil stabilization because PCBs are extremely difficult to degrade and cause secondary pollution of the soil environment and groundwater. Both methods waste a large amount of recyclable resources [8]. Therefore, if it can be reasonably recycled, it can protect the environment and promote the sustainable development of the economy and society [9]. In addition, WPCBs are rich in diverse valuable metal resources such as Cu, Sn, Fe, Ni, Zn, Ag, and Au. These metals, especially precious metals, are very valuable to non-renewable resources. Therefore, the recovery of metals on WPCBs can bring great economic benefits, so it has always been a great concern over the years. Thus, residual WPCBs have become more important in recent years [10].

Residual WPCBs are usually thermoset composites. At present, the main recovery methods for thermosetting composite materials include physical recovery [11], pyrolytic recovery [12], and chemical solvent recovery [13]. The physical recovery consists of cutting the waste thermosetting composites into particles of different sizes and using them as fillers for secondary use. The pyrolytic recovery method comprises treating the thermosetting composites at a high temperature to recover the fiber. The chemical solvent recovery consists of the thermosetting composites being dissolved in chemical reagents, breaking the chemical crosslinking structure in a relatively mild way, allowing us to obtain oligomers or raw material monomers, and the fillers.

Activated carbon is widely used in various fields, such as environmental protection, medicine, chemical industry, sustainable energy, and so on, because of its huge specific surface area, rich porosity, and good electrochemical properties and adsorption properties [14]. Activated carbon is widely used in the fields of adsorption [15], catalysis [16], carrier [17], and electrode material [18,19]. For example, activated carbon can be used as a purifying agent to remove impurities such as humic acid, odor, chromaticity, organic matter, and odor from natural water sources [20]. In the industrial field, activated carbon can be used to adsorb organic pollutants and heavy metal ions produced by printing and dyeing, electroplating, papermaking, pharmaceutical, and other industries [21]. In terms of waste gas treatment, activated carbon can be used as an environmentally friendly, non-toxic, tasteless indoor air purification agent to effectively remove harmful gases, such as formaldehyde, benzene, ammonia, carbon dioxide, and eliminate unpleasant odors [22].

Since the 20th century, the printing and dyeing industry has been developing rapidly. Although the extensive application of organic dyes has benefited thousands of households, it is accompanied by serious water pollution [23]. In printing and dyeing wastewater, the complex structure of organic dyes determines that they are usually highly stable and difficult to degrade, directly endangering the survival of organisms in the water and the stability of the ecosystem, ultimately endangering human life and health. Therefore, a large number of environmental treatment technologies has emerged to tackle this issue. The wastewater treatment technologies mainly include carbon adsorption, membrane filtration, and photo-catalysis [24]. As a common organic dye, methyl orange, is widely used in the printing and dyeing industry. At the same time, methyl orange is also a toxic substance, which will negatively affect the water ecosystem and cause great pollution to the environment when it is released into the environment in large quantities. Activated carbon has an excellent adsorption capacity due to its highly porous structure and surface area [25]. This enables activated carbon to effectively adsorb and remove dye molecules in the water, including organic dyes, fluorescent dyes, etc., thus enabling water purification and dye wastewater treatment. Activated carbon is a natural material, and it is usually prepared from renewable resources, such as wood, bamboo, and fruit shells, which have a low environmental impact and ecological risk. At the same time, activated carbon does not produce secondary pollution during its use, which is in line with the principles of environmental protection and sustainable development. Therefore, carbon adsorption technology is popular because of its simple operation, low cost, and environmentally friendly source of raw materials.

Therefore, residual WPCBs are recycled to prepare activated carbon. This is an effective way of recycling and reusing thermosetting epoxy resin. It not only reduces the deteriorating environmental problems but also expands the raw material sources for the production of activated carbon, bringing significant economic benefits. Herein, the waste thermosetting epoxy resin was used as a raw material to prepare activated carbon materials via a carbonization/activation process to study the adsorption behavior of class-activated carbon on methyl orange. In this experiment, we chose the temperatures of 600 °C and 800 °C for the carbonization and activation of the raw materials because the decomposition temperature of epoxy resin is usually between 300 °C and 500 °C, and because the epoxy resin can be decomposed efficiently and thoroughly at 600 °C and 800 °C. After carbonization, KOH was used as an activator to enable a more suitable activation process for the raw material.

## 2. Experimental Section

### 2.1. The Preparation of Activated Carbon

The waste strengthened epoxy resin boards coming from the garbage station were first pulverized into a powder with diameter less than 0.25 mm (60 mesh) using a pulverizer (XY-6008 Yongzhou Xiaobao Electrical Appliance Company Limited, Yongzhou, China). Then, 15 g of the powder was placed in a tube furnace and heated to a carbonization temperature of 600 °C at a ramp rate of 10 °C/min and calcined for 2.5 h; then, it was naturally cooled to room temperature under the protection of a N_2_ atmosphere (0.5 L/min) to obtain the carbide. The carbide was mixed and ground with the chemical activator KOH at a ratio of 1:3 (wt/wt) in a mortar and was then heated to an activation temperature of 800 °C at a heat rate of 25 °C/min, kept at this temperature for 1 h, and naturally cooled to room temperature. Afterward, the solid particles in the crucible were taken out for grinding, making the crude product.

The crude product was dissolved in an appropriate amount of deionized water and titrated to a neutral pH with hydrochloric acid solution. Deionized water and ethanol were used to rinse and centrifuge it for 3 to 5 times, respectively, to remove residual impure ions such as K^+^ and Cl^−^. The precipitate was then dried at 60 °C for 12 h and ground to obtain the desired activated carbon.

Since the obtained waste circuit boards were pure epoxy resins and different strengthened epoxy resin boards, the activated carbons obtained were named AC-1 AC-2, AC-3, and AC-4, respectively. AC-1 is a pure epoxy resin board, and AC-2, AC-3, and AC-4 are epoxy resin boards strengthened with polyethersulfone with successively increasing content.

### 2.2. Characterization

The morphology of as-prepared activated carbon was observed by a field-emission SEM instrument (Zeiss SIDMA-300, Jena, Germany) after sputter coating with gold. X-ray data were collected on a XRD (Shimadzu XRD-6000, Kyoto, Japan) with a scanning range of 5–60° and a scanning speed of up to 10 degrees per minute. Fourier transform infrared spectra (FT-IR) were obtained using a Nicolet iS10 infrared spectrometer from Thermo Fisher Scientific, Waltham, MA, USA. The KBr method was used with a wavelength range of 400~4000 cm^−1^.

The methyl orange solution was configured as 0, 20, 40, 60, 80, 100, 120, 140, 160 mg/L, and the standard curve was plotted using a Q6 UV-Vis spectrophotometer manufactured by Metash Instruments, Shanghai, China. The concentration was taken as the horizontal axis and absorbance as the vertical axis to plot the standard curve of methyl orange. The standard curve of methyl orange was plotted with the concentration as the horizontal axis and the absorbance as the vertical axis.

The amount of methyl orange adsorbed by the activated carbon sample was calculated with Equation (1):(1)$$q_{t}=\frac{\left(C_{0}-C_{t}\right)}{m}\times V_{0}$$
where $C_{t}$ (mg/L) is the concentration of methyl orange at time t, $C_{0}$ is the initial concentration of methyl orange (mg/L), m is the mass of activated carbon (g), $V_{0}$ is the volume of the solution (L), and $q_{t}$ is the amount of methyl orange adsorbed at time t (mg/g).

## 3. Result and Discussion

### 3.1. Structural Characterization of Activated Carbon

As can be seen in Figure 1, there are several peaks in the FT-IR spectrum.

The strongest peak at 3441 cm^−1^ corresponds to the O-H functional group, and there are three other peaks with smaller intensities at 1101 cm^−1^, 1384 cm^−1^, and 1631 cm^−1^, which represent the C-O-C functional group of aryl ether, the C-O functional group of carboxylate, and the C=O functional group, respectively. Some of these oxygen-containing functional groups are capable of hydrogen bonding or chemically reacting with methyl orange dye. This helps to increase the affinity of the material for organic dyes and enhance its adsorption.

The four prepared samples were subjected to X-ray diffraction (XRD) analysis to analyze the crystal structures, and the results of the XRD tests are shown in Figure 2.

As can be seen in Figure 2, two diffraction peaks appear on the XRD spectra of the four samples, including a strong diffraction peak at 2θ from 20° to 30° and a broader peak at 2θ = 43°, which are diffraction peaks from the (002) and (101) crystal planes of the graphite structure, respectively. The (002) crystal plane is mainly due to the interconnection of the lamellar graphite layers and parallel stacking, and the (101) crystal plane shows that the samples contain a hexagonal honeycomb carbon structure. The appearance of the two facets suggests that some graphitic carbon was formed in the preparation process, which contributes to the improvement in the adsorption of the material. In addition, the intensity of the diffraction peaks corresponding to the (002) facet represents the degree of graphitization of the sample. It can be seen that the graphitization degree of AC-1 is the lowest and that the graphitization degrees of AC-2, AC-3, and AC-4 increase, which is because AC-1 has the richest surface pore structure and the most defects on the surface of the carbon atoms and it has the lowest degree of graphitization. Increasing the specific surface area of the material can provide more adsorption active sites and help to improve the adsorption properties of the material.

In the industry, the adsorption capacity of activated carbon on small molecule impurities can be reflected by the adsorption iodine value of activated carbon, and we mainly want to recycle the industrial waste circuit boards and obtain the activated carbon through high-temperature calcination for a secondary utilization. So, we carried out an iodine adsorption test on the prepared activated carbon samples, and the test results shown in Figure 3 indicate that our activated carbon had a better adsorption capacity for small molecule impurities.

Furthermore, we can obviously find that the iodine adsorption value decreases gradually from sample AC-1 to AC-4, which indicates that the adsorption effect of sample AC-1 on small molecule impurities is better than in the other samples.

The surface morphology of the epoxy resin-based activated carbon was observed using SEM. From Figure 4a–d, it can be observed that samples AC-2, AC-3, and AC-4 have smooth polygonal irregular plate-like solid surfaces, while sample AC-1 has a rough irregular granular surface. Because polyethersulfone has good acid and alkaline resistance, even at high temperatures, the strengthened epoxy resin is difficult to be etched by potassium hydroxide at high temperatures, resulting in a smooth surface of AC-2, AC-3, and AC-4. Compared with the other samples, the rough surface of sample AC-1 increases the pore volume and improves the pore size structure, which is favorable for providing more adsorption sites to improve the adsorption performance of the sample.

### 3.2. Adsorption Behavior of Methyl Orange on Activated Carbon

The adsorption of methyl orange to activated carbon was explored. It is known that the characteristic wavelength of the methyl orange solution is 460 nm. Thus, the methyl orange solutions with concentrations of 0, 20, 40, 60, 80, 100, 120, 140, and 160 mg/L were configured. A wavelength of 460 nm was adopted, and the standard curve of the methyl orange solution in concentration/absorbance was plotted via linear fitting, as shown in Figure 5a. According to the standard curve, the concentration of methyl orange can be calculated from the absorbance of a solution, and then the adsorption amount can be calculated.

The initial concentration of 100 mL methyl orange was 80 mg/L, and four 50 mg activated carbons were added, respectively, to determine the adsorption capacity of methyl orange at different adsorption times, as shown in Figure 5b.

At the initial stage of adsorption, the rate of adsorption increases and the absorption curve is steep. During the middle stage of the process of adsorption, the growth rate of adsorption slows down with time and gradually stabilizes during the late stages of the process of adsorption. This is because in the initial stage of adsorption, the adsorbent has a high concentration, and the large concentration difference of the adsorbate between the methyl orange solution and the adsorbent produces a good transfer impetus, prompting methyl orange to enter into the adsorption site rapidly. The concentration difference decreases and the continuous possession of the adsorption site by methyl orange results in a decreasing force in the solid–liquid push, with the adsorption rate continuing to decrease. When the adsorption sites on the surface of the activated carbon sample were saturated, the adsorption of methyl orange reached an equilibrium, with methyl orange no longer being adsorbed.

The adsorption behavior of methyl orange was studied using quasi-primary and quasi-secondary kinetic models. The fitted results are shown in Figure 6, and the kinetic parameters are presented in Table 1.

In the whole adsorption process from 0 to 200 min, the R^2^ values all are more than 0.90 (Table 1), which means that the adsorption process conforms to the quasi-primary kinetic model and quasi-secondary kinetic model. We can further find from Figure 7 that in the first 40 min of adsorption, the adsorption behavior is more consistent with the quasi-primary kinetic model, while the adsorption process after 40 min is more consistent with the quasi-secondary kinetic behavior. In the first 40 min of adsorption, there are more adsorption sites and a larger concentration difference of the adsorbate, with the adsorption mainly being affected by the diffusion process and having a faster adsorption process. However, after 40 min, most of the adsorption points on the adsorbent are occupied by the adsorbate, and some electrostatic adsorption process of sharing or transferring electron pairs between the adsorbent and the adsorbate occurs. According to the quasi-secondary kinetic model (Table 1), the equilibrium adsorption amounts (q_e_) from AC-1 to AC-4 are 41.051 mg/g, 31.556 mg/g, 28.121 mg/g, and 23.607 mg/g, respectively. Due to the high-temperature stability as well as acid and alkali resistance of polyethersulfone, the anti-activation of epoxy resin is improved (which makes it difficult to form an effective porous structure in the KOH activation process) and its adsorption capacity is low. Therefore, AC-1, without a toughening agent, exhibits the best adsorption behavior.

Moreover, we also compared the adsorption performance of methyl orange with different materials (Table 2) and found that the activated carbon sample prepared from waste epoxy resin could basically reach the adsorption efficiency of commercial activated carbon in this experiment, and we found that the equilibrium adsorption amount also had a great advantage over other substances.

Adsorption thermodynamics can be used to study the adsorption of gases or liquids on solid surfaces. For AC-1, due to it exhibiting the best adsorption behavior, the adsorption thermodynamic behavior of activated carbon was further studied. A series of 50 mL of methyl orange solution at concentrations of 20 mg/L, 40 mg/L, 60 mg/L, 80 mg/L, and 100 mg/L was put into different flasks, with 30 mg of activated carbon to AC-1 being added at each concentration. At constant temperatures of 30 °C, 40 °C, and 50 °C, the concentration of methyl orange in the solution was measured and the equilibrium adsorption amount was calculated. The adsorption isotherms of the samples are shown in Figure 7.

As can be seen from Figure 7, the shape of the adsorption isotherm is an upwardly convex curve. The equilibrium concentration and equilibrium adsorption capacity of the AC-1 adsorption isotherm increased with the increase in the initial concentration of methyl orange in the solution. The larger slope of the adsorption isotherm at low concentrations indicates that adsorption is more likely to occur, which is attributed to the presence of many unoccupied adsorption active sites on the surface of AC-1. The slope of the adsorption isotherm slows down as the concentration of methyl orange increases and eventually tends to reach an equilibrium.

The most commonly used Langmuir and Freundlich isothermal adsorption models [30] were fitted for the isothermal adsorption of methyl orange using AC-1 at 30 °C, 40 °C, and 50 °C. The fitted results are shown in Figure 8 and Table 3.

It can be seen that the correlation coefficients (R_2_) of both the Langmuir and Freundlich models are greater than 0.9, which represents that the fitting effects are significant. As shown in Table 3, the maximum adsorption amounts of (*q**_m_*) for methyl orange using AC-1 were 23.137 mg/g, 30.358 mg/g, and 37.202 mg/g at 30 °C, 40 °C, and 50 °C, respectively. Therefore, the effect of temperature on the adsorption amount exhibited a positive correlation, which indicates that heating can promote the adsorption process. It may be because the pore diameter of the sample surface increased and because the adsorption activity of the pore increased with the increase in temperature. From the n_f_ of the Freundlich adsorption model, it was found that 1/n was equal to 0.83, 0.82, and 0.80—between 0 and 2—representing that absorption can easily occur.

The allocation factor (*R_L_*) is an important property of Langmuir isotherms for describing the affinity of an adsorbate to an adsorbent, and it can be calculated with Equation (2) [31], as shown in Figure 9a.
(2)$$R_{L}=\frac{1}{1+K_{L}C_{0}}$$
where *C*_0_ is the initial concentration before adsorption.

*R_L_* is favorable for adsorption when it is between 0 and 1, unfavorable for adsorption when the partition factor is greater than 1, and unfavorable for linear adsorption when the partition factor is equal to 1 [32]. From Figure 9a, it can be seen that *R**_L_* is between 0 and 1, indicating that the adsorption of AC-1 on methyl orange is favorable.

The reversibility and spontaneity of the adsorption process can be evaluated by calculating the thermodynamic parameters, which include the Gibbs free energy (∆G^0^), enthalpy value (∆H^0^), and entropy value (∆S^0^), which can be calculated with Equation (3) [31] and are fitted as shown in Figure 9b.
(3)$$\Delta G^{0}=-RTlnK_{L}$$
where K_L_ is Langmuir adsorption constant (L/mg), T is the absolute temperature (K), and R is the gas molar constant with a value of 8.314 J/(K·mol).

The curves were plotted with T as the horizontal axis and ∆G_0_ as the vertical axis, and they were fitted linearly, as shown in Figure 9b. The slope of the fitted straight line is −∆S^0^, the intercept is ∆H^0^, and the related parameters are shown in Table 4.

As shown in Table 4, ∆S^0^ is positive, which means that methyl orange is widely distributed on the surface of AC-1.

The increase in randomness of the interface with increasing temperature also indicates that the adsorption process can easily occur. The temperature increases from 303 K to 323 K and ∆G^0^ decreases from 4.85 KJ/mol to 3.98 KJ/mol, indicating that temperature promotes the adsorption process. ∆H^0^ > 0 indicates that this adsorption process absorbs heat from the outside. In addition, it is known that the enthalpy of physical adsorption is less than 20 KJ/mol, the enthalpy of electrostatic adsorption is in the range of 20–80 KJ/mol, and the enthalpy of typical chemical adsorption is in the range of 80–450 KJ/mol [33]. The enthalpy of the adsorption here is 18.09 KJ/mol, so the adsorption in this experiment is mainly physical adsorption.

## 4. Conclusions

In this work, activated carbon with good adsorption properties was prepared from waste epoxy resin boards from used circuit boards with a simple preparation process comprising stepwise carbonization and activation. The prepared activated carbon had rough and porous surface morphology and displayed rich information pertaining its chemical functional groups. The adsorption characteristics and mechanisms of activated carbon on the methyl orange solution were investigated via adsorption thermodynamic and kinetic analyses, which showed that activated carbon had a good adsorption capacity for methyl orange, especially for sample AC-1. The adsorption process is both a physical adsorption and an endothermic process. The preparation of activated carbon with an excellent adsorption effect through the recycling of waste circuit boards not only greatly solves the problem of recycling waste epoxy resin but also holds substantial significance in solving the issue of dyes contaminating soil and water. Additionally, it relevant in the development of a sustainable economy.

## Figures and Tables

**Figure 1 polymers-16-01648-f001:**
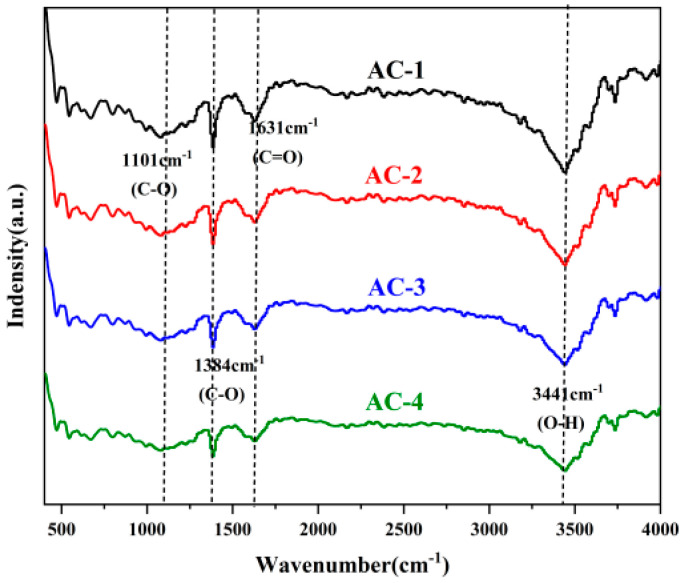
FTIR of AC-1, AC-2, AC-3, and AC-4.

**Figure 2 polymers-16-01648-f002:**
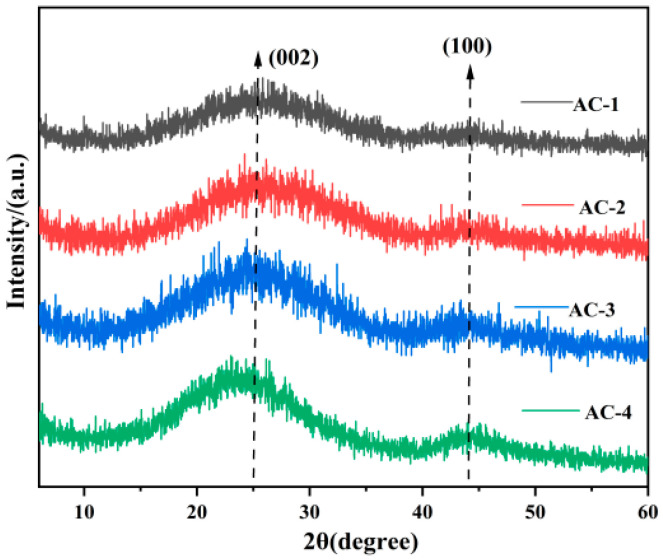
XRD spectra of AC-1, AC-2, AC-3, and AC-4.

**Figure 3 polymers-16-01648-f003:**
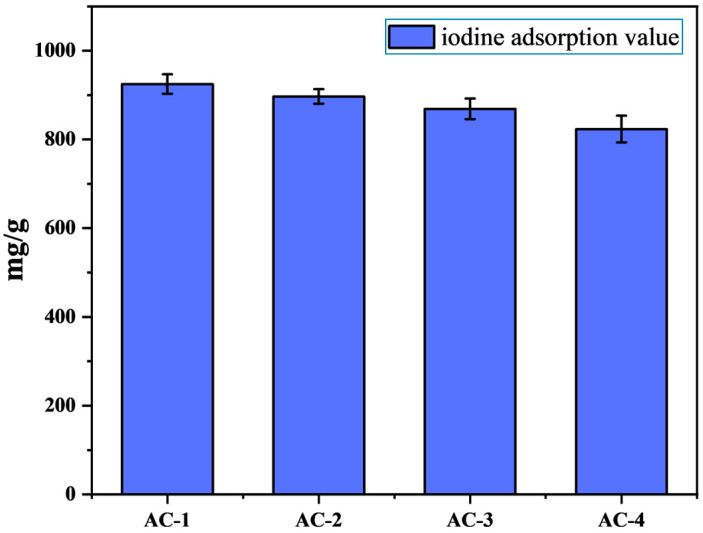
Iodine adsorption values for samples AC-1, AC-2, AC-3, and AC-4.

**Figure 4 polymers-16-01648-f004:**
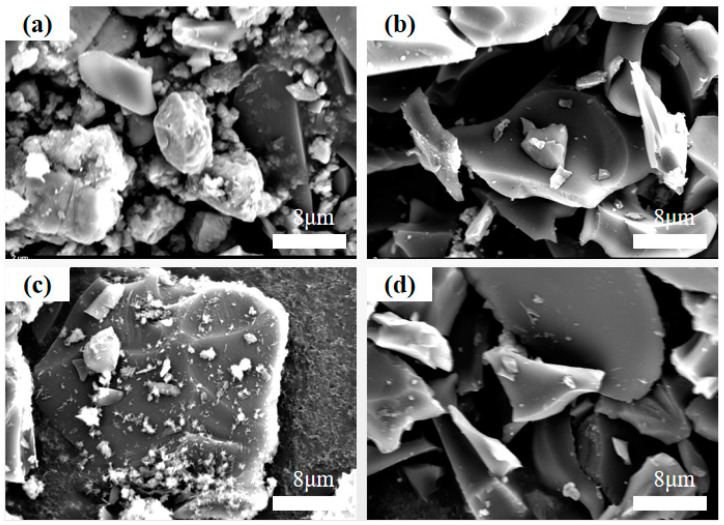
SEM images of (**a**) AC-1, (**b**) AC-2, (**c**) AC-3, and (**d**) AC-4.

**Figure 5 polymers-16-01648-f005:**
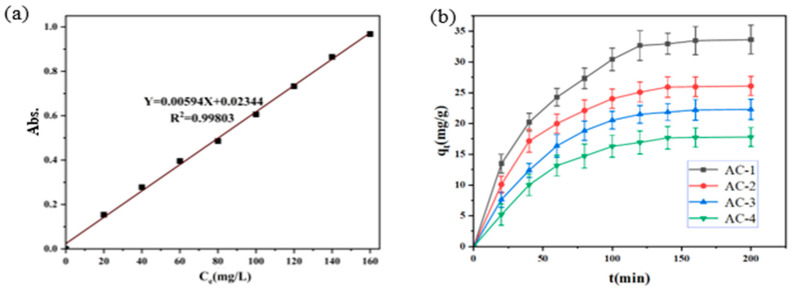
(**a**) Standard curve of methyl orange solution. (**b**) Adsorption of adsorbed methyl orange for AC-1, (**b**) AC-2, AC-3, and AC-4.

**Figure 6 polymers-16-01648-f006:**
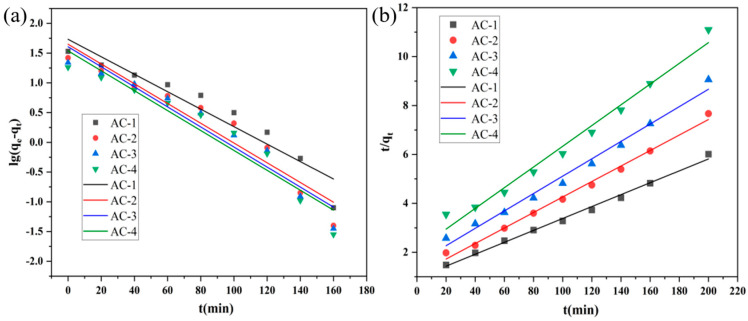
(**a**) Linear fitting using a quasi-primary kinetic model. (**b**) Linear fitting using a quasi-secondary kinetic model of AC-1.

**Figure 7 polymers-16-01648-f007:**
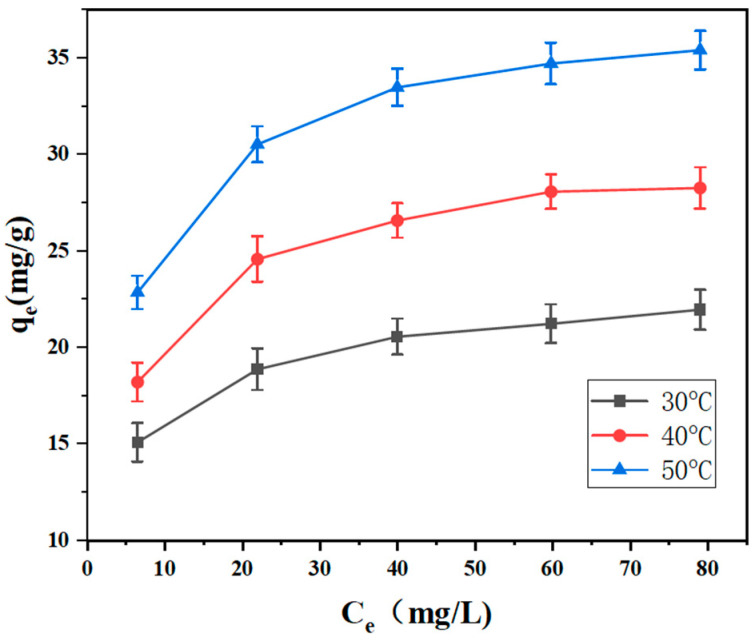
AC-1 adsorption isotherm for methyl orange.

**Figure 8 polymers-16-01648-f008:**
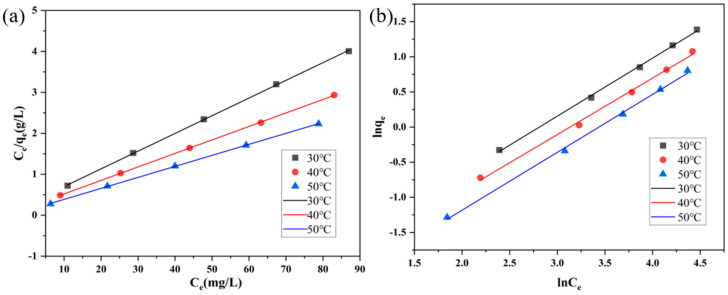
(**a**) Linear fitting according to Langmuir isothermal model. (**b**) Linear fitting using Freundlich isotherm model.

**Figure 9 polymers-16-01648-f009:**
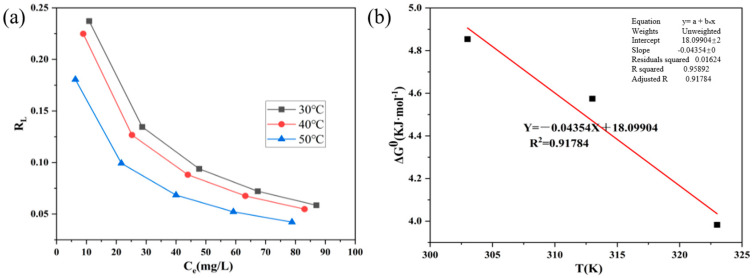
(**a**) Variation of the partition factor with equilibrium concentration. (**b**) Variation of the Gibbs free energy with temperature.

**Table 1 polymers-16-01648-t001:** The adsorption kinetic models’ parameters for the adsorption process.

Samples	Quasi-Primary Kinetic Model			Quasi-Secondary Kinetic Model		
	R^2^	q_e_ mg·g^−1^	k_1_	R^2^	q_e_ mg·g^−1^	k_2_ × 10^−3^
AC-1	0.906	54.250	0.03381	0.993	41.051	0.63
AC-2	0.918	44.646	0.03825	0.993	31.556	0.92
AC-3	0.932	40.509	0.03882	0.984	28.121	0.81
AC-4	0.915	34.531	0.03855	0.978	23.607	0.85

**Table 2 polymers-16-01648-t002:** Adsorption parameters for kinetic modeling of different materials’ adsorption.

Samples	Quasi-Primary Kinetic Model			Quasi-Secondary Kinetic Model			
	R^2^	q_e_ mg·g^−1^	k_1_	R^2^	q_e_ mg·g^−1^	k_2_ × 10^−3^	
AC-1	0.906	54.250	0.03381	0.993	41.051	0.63	
AC	0.9564	44.950	−0.006	0.9992	100	46	[26]
C-CA	0.904	11.004	0.0491	0.97	96.15	0.0006	[27]
CMC	0.392	43.85	0.00084	0.999	38.61	0.0093	[28]
ZnO@AC	0.789	49.329	0.341	0.905	50.853	0.018	[29]

**Table 3 polymers-16-01648-t003:** Fitting parameters of isothermal adsorption model of AC-1.

T/K	Langmuir			Freundlich		
	$q_{m}$ mg·g^−1^	$K_{L}$	$R^{2}$	$n_{F}$	$K_{F}$	$R^{2}$
303K	23.137	0.1608	0.9997	1.2109	0.0981	0.9887
313K	30.358	0.1724	0.9999	1.2137	0.0806	0.9859
333K	37.202	0.2269	0.9998	1.2439	0.05893	0.9877

**Table 4 polymers-16-01648-t004:** Thermodynamic parameters of AC-1.

$∆H^{0}$ (KJ·mol^−1^)	$∆S^{0}$ (J·K^−1^·mol^−1^)	$∆G^{0}$		
		303 K	313 K	333 K
18.09	43.54	4.85	4.57	3.98

## Data Availability

All data are contained within this article.

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
