# Peer review of "Activated Carbon Based on Recycled Epoxy Boards and Their Adsorption toward Methyl Orange"

_polymers, 2024, doi:10.3390/polym16121648_

Round 1
Reviewer 1 Report
Comments and Suggestions for Authors
1. The title of the paper draft is confusing and need revision.
2. Authors provided introduction of printing industry but did not provide the review of literature on the issues related to this industry. Please elaborate the research gap and significance of your work.
3. The XRD spectra of the products need refinement. Two major peaks in XRD spectra are spreading around 2theta of 20 to 30 and 30 to 50, showing large peak width. I think it is not possible, please redo XRD analysis by generating new spectra.
4. Figure 4b and 6, please add error analysis.
5. Please provide a review of literature in comparison to your work.
6. Conclusion is very general and did not provide a quantitative overview of the work.
Author Response
1.The title of the paper draft is confusing and need revision.
Reply: the title is revised as “Activated carbon based on recycled epoxy boards and their adsorption toward methyl orange”
- Authors provided introduction of printing industry but did not provide the review of literature on the issues related to this industry. Please elaborate the research gap and significance of your work
Reply: we have introduced the development of issues related to this industry in introduction.
- The XRD spectra of the products need refinement. Two major peaks in XRD spectra are spreading around 2theta of 20 to 30 and 30 to 50, showing large peak width. l think it is not possible, please redo XRD analysis by generating new spectra.
Reply: We have done XRD for the sample several times, all samples have the similar results. We thought that it may be due to the different types of epoxy resins and toughening agents used in the system which have different degrees of high temperature tolerance. In the later calcination process, amorphous activated carbon is generated instead of crystalline carbon materials to form the broad and weak G and D peaks.
- Figure 4b and 6, please add error analysis
Reply: We have add the error analysis in Figure 4b and 6.
- Please provide a review of literature in comparison to your work
Reply: We have added a table to compare our work with the data of literatures.
- Conclusion is very general and did not provide a quantitative overview of the work
Reply: We rewrote our conclusions to summarize our work.

Reviewer 2 Report
Comments and Suggestions for Authors
Abstract:
1. Clarify "stepwise carbonization-activation methods" specifics.
2. Detail surface characterization techniques used for AC.
3. Expand on "multiple oxygen-containing groups" identification.
4. Justify the selection of methyl orange for adsorption studies.
5. Compare adsorption data with existing literature values.
6. Elaborate on economic benefit analysis methodology.
Section 1:
1. Specify the "considerable difficulty" in recycling epoxy resins.
2. Detail "secondary pollution" types from incineration and landfill.
3. Clarify how activated carbon promotes sustainable economy.
4. Justify activated carbon's selection for dye adsorption over alternatives.
5. Expand on "significant economic benefits" of recycling PCBs for AC.
6. Consider citing https://doi.org/10.1016/j.compstruct.2020.112967
Section 2:
1. Justify "600 °C" and "800 °C" as chosen temperatures.
2. Explain KOH ratio choice for activation process.
3. Clarify criteria for "appropriate amount" of deionized water.
4. Detail gold sputter coating necessity for SEM imaging.
5. Specify "appropriate pH value" for methylene orange solution.
6. Address potential variability in activated carbon samples AC-1 to AC-4.
Section 3:
1. Clarify how O-H functional group enhances dye adsorption.
2. Justify "graphitization degree" impact on adsorption efficiency.
3. Detail SEM methodology for surface morphology analysis.
4. Explain methyl orange concentration selection for experiments.
5. Compare kinetic models' fit beyond R2 values for robustness.
6. Discuss physical adsorption's implications on recycling efficacy.
Section 4:
1. Specify "simple feasible preparation process" steps more clearly.
2. Detail how temperature influences adsorption capacity mechanistically.
3. Clarify link between surface morphology and adsorption efficiency.
Author Response
Abstract:
- Clarify "stepwise carbonization-activation methods" specifics.
Reply: The step-by-step carbonization-activation method divides the carbonization and activation of the raw material into two steps, firstly, the raw material is carbonized at 600oC, and then the carbonized raw material is activated at 800oC after adding activator.
- Detail surface characterization techniques used for AC
Reply: The morphology of the activated carbon samples prepared after sputter plating of gold was observed by field emission scanning electron microscopy (SIDMA-300).
- Expand on "multiple oxygen-containing groups" identification.
Reply: The FTIR spectra show several distinct peaks of oxygen-containing groups, the strongest peak at 3441 cm-1 corresponds to the O-H functional group, and there are three other peaks with smaller intensity at 1101 cm-1, 1384 cm-1 and 1631 cm-1, which represent the C-O-C functional group of the aryl ether, the C-O functional group of carboxylate, and the C=O functional group of aryl ethers, C-O functional group of carboxylates and C=O functional group, respectively.
- Justify the selection of methyl orange for adsorption studies
Reply: As a common organic dye, methyl orange is widely used in printing and dyeing industry, but at the same time, methyl orange is also a toxic substance, which will negatively affect the water ecosystem and cause great pollution to the environment when it is released into the environment in large quantities. At the same time, Methyl Orange has very obvious spectral characteristics, which can be easily determined by spectrophotometer, which makes it very convenient for adsorption studies and performance evaluation under laboratory conditions.
- Compare adsorption data with existing literature values.
Reply: We have added a table comparing our work with literature data in table2.
- Elaborate on economic benefit analysis methodology.
Reply: Activated carbon can convert hazardous substances into recyclable resources through adsorption, separation or conversion. In our research, we can recycle waste circuit boards, which are difficult to recycle, to prepare activated carbon. Through the treatment of waste and recycling of resources, we can not only reduce the waste of resources and pollution to the environment, promote the recycling of resources, but also realize the sustainable development of the economy.
Section 1:
- Specify the "considerable difficulty" in recycling epoxy resins.
Reply: Epoxy resins are three-dimensional cross-linked polymers with a complex chemical structure, usually consisting of epoxy groups and polyanhydrides containing aromatic or aliphatic structures, among others. This structure is difficult to be decomposed effectively in the recycling process, and its insoluble and insoluble properties make it difficult to be recycled. Secondly, the epoxy resin itself has excellent properties such as high heat resistance and chemical resistance, which also makes it difficult to recycle the epoxy resin in the recycling process.
- Detail "secondary pollution" types from incineration and landfill.
Reply: Incineration produces polluting gases that cause air pollution and jeopardize human health. Direct landfill is not conducive to soil stabilization because PCBs are extremely difficult to degrade and will cause secondary pollution to the soil environment and groundwater.
- Clarify how activated carbon promotes sustainable economy.
Reply: Activated carbon is widely used in water treatment, atmospheric purification, soil remediation and other fields. Activated carbon can transform harmful substances into recyclable resources through adsorption, separation or transformation. Through the treatment of waste and the recycling of resources, activated carbon helps to reduce the waste of resources and pollution of the environment, promote the recycling of resources, and realize the sustainable development of the economy.
- Justify activated carbon's selection for dye adsorption over alternatives
Reply: Activated carbon has a high pore structure and surface area, resulting in an extremely high adsorption capacity. This enables activated carbon to effectively adsorb and remove dye molecules in water, including organic dyes, fluorescent dyes, etc., thus enabling water purification and dye wastewater treatment. Activated carbon is a natural material, usually prepared from renewable resources such as wood, bamboo and fruit shells, which has a low environmental impact and low ecological risk. At the same time, activated carbon does not produce secondary pollution during its use, which is in line with the principles of environmental protection and sustainable development.
- Expand on "significant economic benefits" of recycling PCBs for AC
Reply: Waste PCB is a toxic substance for, and its use in the production of activated carbon not only solves the problem of pollution, but also realizes the recycling and reuse of resources, which reduces the cost of producing activated carbon and improves the economic efficiency while reducing pollution.
- Consider citing https://doi.org/10.1016/compstruct.2020.112967
Reply: We have cited this article.
Section 2:
- Justify "600 °C" and "800 °C" as chosen temperatures.
Reply: The decomposition temperature of epoxy resin is usually between 300oC and 500oC. In this experiment, we chose 600oC and 800oC to carbonize and activate the raw materials step by step, mainly because these two temperatures basically cover the main temperature range of the decomposition of epoxy resin, which can realize the efficient and thorough decomposition of epoxy resin.
- Explain KOH ratio choice for activation process.
Reply: The use of KOH reacts with the carbon source at high temperatures to generate activated carbon and helps to increase the surface area and adsorption properties of the activated carbon. The one to three ratio allows for a moderate activation process without destroying the structure and stability of the activated carbon due to excess activator.
- Clarify criteria for "appropriate amount" of deionized water.
Reply: Just add appropriate amount of deionized water and prepare the sample into a homogeneous suspension. And ensure the balance of the centrifuge during centrifugation.
- Detail gold sputter coating necessity for SEM imaging.
Reply: A metallic gold coating is applied to the surface of the prepared sample using a sputter coater to form a stable and uniform conductive layer on the surface of the sample, which enhances the conductivity and results in a better scanned image.
- Specify “appropriate pH value" for methylene orange solution.
Reply: Selection of appropriate PH value can change the adsorption interactions between methyl orange molecules and activated carbon surface. The adsorption was tested in the experiment by adjusting the PH to an acidic environment.
- Address potential variability in activated carbon samples AC-1 to AC-4
Reply: The activated carbon samples AC-1 to AC- were kept as consistent as possible during the preparation of the operational conditioning to ensure its single variable principle. The consistency of conditions was ensured as much as possible.
Section 3:
- Clarify how O-H functional group enhances dye adsorption.
Reply: Oxygen-containing functional groups such as O-H are capable of hydrogen-bonding or chemically reacting with the methyl orange dye thereby enhancing the adsorption of the dye.
- Justify "graphitization degree" impact on adsorption efficiency.
Reply: The low degree of graphitization and rich surface pore structure increase the specific surface area of the material to provide more adsorption active sites, which helps to improve the adsorption of the material.
- Detail SEM methodology for surface morphology analysis.
Reply: The activated carbon samples obtained from the preparation were prepared in appropriate shapes and sizes, and the surface of the prepared samples were coated with conductive coating to form a conductive layer on the surface (metallic gold was used as the conductive coating in the experiments), and then the morphology of the sample surfaces was observed by scanning electron microscopy under vacuum conditions.
- Explain methyl orange concentration selection for experiments.
Reply: In the experiment we prepared several concentrations of methyl orange solution and the concentration gradient was increased sequentially to find the initial concentration with the best adsorption effect for kinetic simulation and linear fitting.
- Compare kinetic models' fit beyond R2 values for robustness.
Reply: The value of the regression equation R2 was 0.99803, which is very close to 1, indicating a very high degree of fit and good robustness
- Discuss physical adsorption's implications on recycling efficacy.
Reply: Physical adsorption enhances the adsorption of dyes to a certain extent, but in some high concentrations, physical adsorption may also lead to adsorption saturation too quickly, resulting in a shorter regeneration cycle of the adsorbent, in addition to specific adsorbent selectivity ken be disturbed.
Section 4.
- Specify "simple feasible preparation process" steps more clearly.
Reply: The preparation process was divided into two steps, carbonization and activation, to reduce the tediousness of the preparation process. In the preparation, carbonization was carried out at 600oC, and then the product obtained from carbonization was activated at 800oC after adding activator, and the sample obtained was centrifuged and washed to obtain the sample activated carbon with good adsorption effect.
- Detail how temperature influences adsorption capacity mechanistically
Reply: Temperature affects the adsorption capacity of activated carbon through various factors such as physical adsorption mechanism, diffusion rate of dyes and chemical properties. The most important of these is that temperature increases the thermal movement and kinetic energy of molecules, thus accelerating the diffusion rate of the target substance. This increases the likelihood that the substance will reach the adsorption site of the activated carbon, and therefore the rate of adsorption of the activated carbon may increase as the temperature increases, thus affecting the adsorption capacity.
- Clarify link between surface morphology and adsorption efficiency.
Reply: It is shown that the rougher morphology increases the pore capacity and improves the pore structure of the activated carbon, which can provide more adsorption sites to improve the adsorption properties of the sample, thus increasing the adsorption rate of the sample.

Reviewer 3 Report
Comments and Suggestions for Authors
Report on the manuscript polymers-2925014-peer-review-v1- entitled “Activated carbon based on discarded epoxy resin and their adsorption toward methyl orange”. The submitted manuscript should be revised. This work represented preparation of activated carbon (AC) through stepwise carbonization-activation methods followed by investigating the adsorption of methyl orange. In short, the following points should be addressed:
1. The language of the manuscript should be checked.
2. BET analysis is a key analysis for the prepared activated carbon materials, please, try to do it?
3. “160mg/L”, there is a distance between the number and unit, please, check all manuscript parts for this error.
4. Poor R2 (0.917) in figure 8B, why?
5. The conclusion should have the main achievement of the manuscript with the found results not general words.
Comments on the Quality of English Language
It should be revised.
Author Response
Report on the manuscript polymers-2925014-peer-review-v1- entitled “Activated carbon based on discarded epoxy resin and their adsorption toward methyl orange". The submitted manuscript should be revised. This work represented preparation of activated carbon (AC) through stepwise carbonization-activation methods followed by investigating the adsorption of methyl orange. In short, the following points should be addressed:
- The language of the manuscript should be checked.
Reply: We have carefully checked and revised our language.
- BET analysis is a key analysis for the prepared activated carbon materials, please, try to do it?
Reply: Thank you for your valuable suggestion. For porous materials, porosity and specific surface area analysis is an indispensable and important step. Usually, N2 isothermal adsorption and desorption curve (commonly known as BET method) is used to determine the specific surface area, pore volume, pore size distribution, etc.
However, this article “How Reproducible are Surface Areas Calculated from the BET Equation?” reported that when testing materials containing micropores, the results obtained by simply applying the classic BET test method will be significantly different. Often the same sample, do two or three BET tests, the results are very different, sometimes even have an order of magnitude difference.
For this work, we mainly want to recycle the waste circuit board to obtain activated carbon by high-temperature calcination for secondary utilization. In the industry, In the industry, the adsorption capacity of activated carbon for small molecular impurities can be reflected by the adsorpted iodine value of activated carbon. The iodine adsorption values of AC-1, AC-2, AC-3, AC-4 are 925 mg/g, 897 mg/g, 846 mg/g, 822 mg/g, respectively, indicating that our activated carbon had a good ability to adsorb small molecules.
It shows that our activated carbon has a good ability to adsorb small molecules
- “160mg/L", there is a distance between the number and unit, please, check all manuscript parts for this error.
Reply: We are terribly sorry. We revised this low-level error and checked the full text to correct similar errors
- Poor R2 (0 .917) in figure 8B, why?
Reply: The activated carbon adsorption process may be affected by several factors, such as temperature, pH, and particle size. All of these factors may not be modeled well enough to account for variations in the data, resulting in a poor fit. In this case temperature is a very important factor.
- The conclusion should have the main achievement of the manuscript with the found results not general words.
Reply: We rewrote our conclusions to summarize our work.

Round 2
Reviewer 2 Report
Comments and Suggestions for Authors
All major comments were adequately addressed and the Authors have done an admirable job of improving the quality of the manuscript. Therefore, it can be accepted without any structural modification.
Reviewer 3 Report
Comments and Suggestions for Authors
Accepted